# Urban Air Pollution Particulates Suppress Human T-Cell Responses to *Mycobacterium Tuberculosis*

**DOI:** 10.3390/ijerph16214112

**Published:** 2019-10-25

**Authors:** Olufunmilola Ibironke, Claudia Carranza, Srijata Sarkar, Martha Torres, Hyejeong Theresa Choi, Joyce Nwoko, Kathleen Black, Raul Quintana-Belmares, Álvaro Osornio-Vargas, Pamela Ohman-Strickland, Stephan Schwander

**Affiliations:** 1Physiology and Integrative Biology, Rutgers University, Piscataway, NJ 08854, USA; oai5@gsbs.rutgers.edu; 2Department of Microbiology, National Institute of Respiratory Diseases (INER), Mexico City 1408, Mexico; carranza.salazar.claudia@gmail.com (C.C.); marthatorres98@yahoo.com (M.T.); 3Environmental and Occupational Health Sciences Institute, Rutgers, Piscataway, NJ 08854, USA; sarkarsr@sph.rutgers.edu (S.S.); na1004ya@gmail.com (H.T.C.); kgb3@eohsi.rutgers.edu (K.B.); 4Department of Environmental and Occupational Health, Rutgers School of Public Health, Piscataway, NJ 08854, USA; nonyenwa55@gmail.com; 5Instituto Nacional de Cancerología, Mexico City 1408, Mexico; qbro@hotmail.com; 6Department of Pediatrics, University of Alberta, Edmonton, AB T6G 1C9, Canada; osornio@ualberta.ca; 7Department of Biostatistics Rutgers University School of Public Health, Piscataway, NJ 08854, USA; ohmanpa@sph.rutgers.edu; 8Department of Urban-Global Public Health, Rutgers University School of Public Health, Newark, NJ 07102, USA

**Keywords:** M.tb, *PM_2.5_*, immunity, proinflammatory cytokines, T-bet

## Abstract

Tuberculosis (TB) and air pollution both contribute significantly to the global burden of disease. Epidemiological studies show that exposure to household and urban air pollution increase the risk of new infections with *Mycobacterium tuberculosis* (M.tb) and the development of TB in persons infected with *M.tb* and alter treatment outcomes. There is increasing evidence that particulate matter (PM) exposure weakens protective antimycobacterial host immunity. Mechanisms by which exposure to urban PM may adversely affect *M.tb*-specific human T cell functions have not been studied. We, therefore, explored the effects of urban air pollution *PM_2.5_* (aerodynamic diameters ≤2.5µm) on M.tb-specific T cell functions in human peripheral blood mononuclear cells (PBMC). *PM_2.5_* exposure decreased the capacity of PBMC to control the growth of M.tb and the M.tb-induced expression of CD69, an early surface activation marker expressed on CD3^+^ T cells. *PM_2.5_* exposure also decreased the production of IFN-γ in CD3^+^, TNF-α in CD3^+^ and CD14^+^ M.tb-infected PBMC, and the M.tb-induced expression of T-box transcription factor TBX21 (T-bet). In contrast, *PM_2.5_* exposure increased the expression of anti-inflammatory cytokine IL-10 in CD3^+^ and CD14^+^ PBMC. Taken together, *PM_2.5_* exposure of PBMC prior to infection with M.tb impairs critical antimycobacterial T cell immune functions.

## 1. Introduction

The control of the human infection with *Mycobacterium tuberculosis* (M.tb), the causative agent of tuberculosis (TB), requires innate and adaptive (T cell-dependent) antimycobacterial immune responses [1]. Protective human host immunity against M.tb is primarily cell-mediated, and involves Th1 immunity [2] with production of interferon- γ (IFN-γ) [3] and tumor necrosis factor- α (TNF-α) [4]. 

Integral effector functions of T cells during M.tb infection include the production of IFN-γ and the lysis of M.tb-infected phagocytes [5]. TNF-α production upon M.tb infection of human blood monocytes [6] and T cells [4] in vitro plays a vital role in protective host immunity against M.tb, and, in synergy with IFN-γ, is required for mycobacterial growth control [7] and optimal macrophage activation [8]. Conversely, anti-inflammatory cytokine interleukin-10 (IL-10) dampens Th1 cell responses to M.tb infection, T cell proliferation [9] and IFN-γ production [10]. Furthermore, IL-10 promotes M.tb survival and higher levels of IL-10 are positively correlated with the severity of the clinical phenotype of TB [11]. Multiple clinical conditions such as HIV infection [12], malnutrition [13], long-term corticosteroid therapies and antineoplastic chemotherapies [14] and TNF inhibitors [15], facilitate development and progression of TB providing further evidence for the requirement of intact T cell immunity for protective host immunity against M.tb.

Recent studies have demonstrated that exposure to cigarette smoke weakens M.tb-induced pulmonary T cell responses [16], that household air pollution exposure facilitates the development of active TB [17] and that exposure to urban air pollution adversely affects anti-tuberculous treatment outcomes [18]. 

In earlier studies, we have shown in peripheral blood mononuclear cells (PBMC) that diesel exhaust particles (DEP), a component of urban outdoor PM, alter M.tb-induced inflammatory cytokine and IRF-1 and NF-ĸB target gene expression in a dose-dependent manner [19]. We also reported that exposure to urban air pollution *PM_2.5_* and PM_10_ (particulate matter with aerodynamic diameters ≤2.5µm and 10µm, respectively) decreases the expression of human β-defensins 2 and 3 (HBD-2 and HBD-3) upon infection with M.tb and induces cellular senescence leading to increased intracellular M.tb growth in A549 cells [20]. In a recent study we have further shown that impairment of innate and adaptive antimycobacterial immune functions of human bronchoalveolar cells and PBMC correlate with the PM load in the autologous alveolar macrophages [21].

Studies assessing the effects of PM on T-cell immunity are lacking to date. The current study therefore assessed whether *PM_2.5_* exposure in vitro impacts human peripheral blood T cell responses to M.tb. 

## 2. Materials and Methods 

### 2.1. Study Approval

This study was conducted in accordance with the Declaration of Helsinki, and protocol was approved by the Institutional Review Board of Rutgers, The State University of New Jersey in Newark and New Brunswick (protocol number 2012001383). All study subjects provided signed written informed consent prior to any study interactions.

### 2.2. Human Subjects

A total of 21 volunteers (fourteen females and seven males, median age 28 years, range 20–62 years) was recruited from students and staff at Rutgers University and the local community to provide blood samples for the various experiments. A total of 100 mL heparinized, peripheral blood was obtained by venipuncture from each of the study participants. Persons undergoing long-term medications, smokers, or users of illicit drugs were excluded from study participation. 

### 2.3. Preparation of PBMC

PBMC were isolated from heparinized whole peripheral venous blood by Ficoll gradient centrifugation as previously described [22]. Briefly, blood was diluted with complete culture medium at a 1:1 volume ratio, overlaid on Ficoll-Paque and subjected to gradient density centrifugation (1200 rpm at 21 °C for 45 min). PBMC were obtained from the gradient interface, washed three times in RPMI 1640, re-suspended in complete culture medium, counted in a hemocytometer and adjusted to required concentrations for the various experiments. Viability of PBMC was 98–100% by trypan blue exclusion in all experiments.

### 2.4. Collection of PM_2.5_ and Preparation for in Vitro Exposure of PBMC 

Urban *PM_2.5_* was collected in the context of the NIEHS-funded project entitled ‘Air Pollution Particle Effects on Human Antimycobacterial Immunity’ (5R01ES020382, PI S. Schwander) on the rooftop of the National Institute of Ecology and Climate Change (Instituto Nacional de Ecología y Cambio Climático (INECC)) in the Iztapalapa municipality of Mexico City. *PM_2.5_* was collected with high-volume samplers (GMW Model 1200, VFC HVPM10, airflow rate 1.13 m^3^/min) on nitrocellulose filters, in 2012/2013, as previously described [14]. Following removal from the nitrocellulose filters [20], *PM_2.5_* was pooled and weighed using a semi-micro balance (CPA225D; Sartorius, Bohemia, NY, USA) and stored at 4 °C in baked glass flasks until use. Stock suspensions of autoclaved *PM_2.5_* (1 mg/mL) were prepared by sonication (5 min; ultrasonic cleaner model 3510R-DTH; Branson, Danbury, CT, USA) in complete culture medium (RPMI 1640 (BioWhittaker, Lonza Walkersville, MD, USA) supplemented with L-glutamine (Thermo Fisher, Waltham, MA, USA) and 10% pooled human AB serum (Valley Biomedical, Inc., Winchester, VA, USA)) and further diluted to final concentrations of 1 and 5 µg/mL prior to in vitro exposure of PBMC. 

### 2.5. Preparation of M.tb for in Vitro Infection

Preparation of M.tb (H37Ra, ATCC 25177, Manassas, VA, USA) for PBMC infection was done as described previously [22]. M.tb suspensions were prepared in Middlebrook 7H9 broth medium supplemented with 10% albumin dextrose catalase (BD Bioscience) and 0.2% glycerol. After a 21-day incubation period at 37 °C on an orbital shaker, M.tb stock suspensions were harvested and concentrations assessed by colony-forming unit (cfu) counts on 7H10 solid agar plates after 21-day incubations. Aliquots were then made and stored at −86 °C until use in in vitro infection experiments. 

For PBMC infection experiments, single cell M.tb suspensions were prepared as follows: frozen M.tb stock was thawed, centrifuged for 5 min at 8000× *g* and re-suspended in complete culture medium. Single bacterial cell suspensions were generated by declumping (5 min. vortexing with 5 sterile 3-mm glass beads) from M.tb stock suspensions. An additional centrifugation step (350× *g* for 5 min.) was added to remove any remaining M.tb clumps. To generate desired multiplicities of infection (MOI, i.e. the ratios of M.tb to monocytes for in vitro infections) of 1 (MOI1) and 5 (MOI5), percentages of monocytes in PBMC from each study participant were assessed by flow cytometry. Concentrations of M.tb in thawed M.tb stock suspensions were confirmed in each infection experiment by cfu assays.

### 2.6. PM_2.5_ Exposures of PBMC 

For each experimental condition, PBMC were seeded in duplicate wells into 96-well round bottom plates (200,000 cells in 100 µL complete culture medium/well, for cfu and LDH assays) or into 5 ml round bottom polypropylene tubes (10^6^ cells/mL complete culture medium, for all flow cytometry and western blot experiments). PBMC in complete culture medium were then either exposed to *PM_2.5_* alone or pre-exposed to *PM_2.5_* for 20 h and then infected with M.tb for an additional 18 h. PBMC were exposed to *PM_2.5_* at final concentrations of 0 (No PM control), 1 and 5 µg/mL and infected with M.tb in complete culture medium and incubated at 37 °C with 4% CO_2_ in a humidified environment. The PM concentrations of 1 and 5 µg/mL (or 0.26 and 1.315 µg/cm^2^ as calculated by area) used in our experiments are considerably lower than the PM concentrations in human airways during inhalation real-world urban air pollution exposures. An average person in Mexico City is calculated to inhale around 80 µg of PM per ml of lining fluid per day [20].

### 2.7. PM_2.5_ Exposures, Infection with M.tb, and CFU Assays

PBMC were exposed to *PM_2.5_* at 1 and 5 µg/mL for 20 h and subsequently infected with M.tb at MOI 1 and 5 (37 °C) for 2 h, washed twice with warm complete culture medium to remove extracellular M.tb, and further incubated at 37 °C in 5% CO_2_ for 1 h (day 0), and for 1, 4, or 7 days, respectively. M.tb growth was assessed as described previously [22]. Briefly, PBMC exposed to *PM_2.5_* only, infected with M.tb only, or *PM_2.5_*-exposed and M.tb-infected were washed twice with 1x phosphate-buffered saline (PBS) and then lysed with 0.1% Sodium Dodecyl Sulfate (SDS) (10 min at room temperature) to release any remaining viable intracellular M.tb. The lysis process was stopped by neutralizing the action of SDS with Middlebrook 7H9 broth enriched with 20% bovine serum albumin (BSA). Four serial cell lysate dilutions (1:10) were then plated in triplicate (10 µl each) onto 7H10 agar plates, incubated at 37 °C for 21 days to allow cfu assessments using a stereomicroscope (40x, Fisher Scientific, Massachusetts USA). 

### 2.8. Lactate Dehydrogenase (LDH) Assay

LDH levels were assessed in PBMC culture supernatants (50 µL) following exposures to *PM_2.5_* at final concentrations of 0 (No-PM control), 1, or 5 µg/mL for 0, 1, 4, and 7 days) and/or infections with M.tb. PBMC culture supernatants were transferred into 96-well assay plates, and 50 µL of substrate (CytoTox 96 Non-radioactive cytotoxicity Assay, Promega, Madison, WI, USA) added to each well. Following incubation at room temperature for 30 min in the dark, stop solution (50 μL) was added to each well and absorbance recorded at 493 nm with an ELISA reader (Thermo Scientific Multiskan FC, Finland). Cellular toxicity (reduced cell viability) was defined in percent (%) LDH leakage (ratio of ODs of *PM_2.5_*-exposed/M.tb-infected PBMC to ODs of unexposed PBMC x 100).

### 2.9. Cell Surface and Intracellular Cytokine Immunostaining 

The expression of T cell surface markers (CD3, CD4, CD8, CD16, CD69), IFN-γ, TNF-α, and IL-10 was assessed in live cells by flow cytometry through the exclusion of dead cells by fixable viability dye eFluor 780 labelling. Uninfected PBMC or *PM_2.5_*-exposed, M.tb-infected, or *PM_2.5_*-pre-exposed (exposure to *PM_2.5_* prior to M.tb infection) and M.tb-infected PBMC (10^6^/mL in 5ml round bottom polypropylene tubes) were washed twice with PBS, re-suspended in flow cytometry staining buffer (Affymetrix eBioscience, San Diego, CA, USA) and stained with fluorescence-conjugated monoclonal antibodies (Affymetrix eBioscience, San Diego, CA except where noted otherwise): anti-Human CD3 (clone OKT3)-Alexa Fluor 700, anti-Human CD4 (clone OKT4)-Alexa Fluor 488, anti-Human CD8a (clone RPA-T8)-PE-Cyanine7, and anti-Human CD16 (clone 3G8)-PE/Dazzle 594 (BioLegend, San Diego, CA, USA) and CD69 (clone FN50)-PE-CF594 (BD Biosciences, San Jose, CA, USA). For intracellular cytokine staining and detection of IFN-γ, TNF-α and IL-10, 2 µl/mL protein transport inhibition cocktail (Brefeldin A and Monensin, Affymetrix eBioscience, San Diego, CA, USA) was added to the respective PBMC cultures during the last 6–10 hours of incubation. PBMC were then washed twice with PBS and fixable viability dye eFluor 780 (Affymetrix eBioscience, San Diego, CA, USA) added followed by surface staining, fixation and addition of permeabilization buffer (Affymetrix eBioscience, San Diego, CA, USA). PBMC were then stained with anti-Human IFN gamma (clone 4S.B3)-phycoerythrin, anti-Human TNF alpha (clone Mab11)-allophycocyanin and anti-Human IL-10 (clone BMS131-2FI)-Fluorescein isothiocyanate and acquired by Gallios flow cytometer (Beckman Coulter 405nm, 488nm, 633nm laser). Data were analyzed with Kaluza Analysis software (Beckman Coulter, Indianapolis, IN, USA).

### 2.10. Flow Cytometry Analysis

A sequential gating strategy was used to analyze IFN-γ, TNF-α, IL-10, and CD69 levels in T cells and monocytes. An initial gate was set in a forward scatter (FSC) vs. side scatter (SSC) dot plot to include cells and exclude debris and cell aggregates. Live cells were then gated on the channel of Fixable Viability dye eFluor 780 to exclude dead cells. Compensation was performed using single color controls prepared from negative control and anti-mouse Ig compensation beads (BD Biosciences, Franklin Lakes, NJ, USA). Logical scaling was used when necessary for compensation using Kaluza Analysis Software. Appropriate and matched isotype controls were used for negative controls and FMO (Fluorescence Minus One) controls were used to distinguish positive from negative cell populations. For each sample, 70,000 live cell gates were created for acquisition of cells. 

### 2.11. Apoptosis Assay by Annexin V Staining

To determine if *PM_2.5_* exposure induces apoptosis of monocytes and lymphocytes in PBMC, uninfected, *PM_2.5_*-exposed, M.tb-infected or *PM_2.5_*-pre-exposed and M.tb-infected PBMC (10^6^ /mL in 5ml round bottom polypropylene tubes) were washed twice with PBS and phosphatidylserine exposure assessed by flow cytometry utilizing a FITC Annexin-V Apoptosis Detection Kit (BD Pharmingen cat: 556547). Annexin-V FITC/propidium iodide double-staining, to assess the proportion of mononuclear cells that were undergoing apoptosis was performed according to manufacturer’s protocol. Induction of apoptosis was evaluated by flow cytometry using a Gallios flow cytometer (Beckman Coulter, Indianapolis, IN, USA) and Kaluza Analysis software (Beckman Coulter, Indianapolis, IN, USA).

### 2.12. Transmission Electron Microscopy 

Transmission electron microscopy (TEM) was employed to examine cellular uptake of *PM_2.5_* and M.tb by monocytes in PBMC. Preparations for transmission electron microscopy (TEM) were done as follows: uninfected, *PM_2.5_*-exposed (5 µg/mL for 20 h), M.tb-infected (18 h) or *PM_2.5_*-pre-exposed (5 µg/mL for 20 h) and M.tb-infected (18 h infection following initial 20 h *PM_2.5_*-exposure) PBMC were fixed in 2.5% glutaraldehyde-4% paraformaldehyde in 0.1 M cacodylate for 1h at room temperature. PBMC were then washed with PBS, post-fixed in buffered 1% osmium tetroxide, dehydrated in a graded series of acetone, and embedded in Epon resin. Fixed cells were cut into 90-nm thin sections using a Leica EM UC6 ultramicrotome, and sectioned grids stained with a saturated solution of uranyl acetate and lead citrate. Images were captured with an AMT XR111 digital camera (Advance Microscopy Techniques, Woburn, MA, USA) on a Philips CM12 transmission electron microscope.

### 2.13. Evaluation of Transcription Factor T-Bet by Western Blot

To examine *PM_2.5_* effects on the expression levels of T-bet, PBMC were exposed to 5 µg/mL of *PM_2.5_* in complete culture media for 20 h followed by M.tb infection at MOI1 or MOI5 or exposed to purified protein derivative (PPD, the antigen Gemisch used in tuberculin skin testing, Statens Serum Institute, Copenhagen, Denmark) at 10 µg/mL. Following an infection period of 18 h, PBMC were lysed with RIPA (radio immunoprecipitation assay) lysis buffer system (Santa Cruz Biotechnology, Dallas, TX) and protein content quantified by Bradford Protein Assay (Bio-Rad laboratories, Hercules, CA). Protein lysates were then analyzed by SDS/PAGE followed by transfer onto polyvinylidene difluoride (PVDF) membranes. T-bet and glyceraldehyde 3-phosphate dehydrogenase (GAPDH)-specific proteins were analyzed by western blotting with specific antibodies (Cell Signaling Technology, Danvers, MA, USA).

### 2.14. Statistical Analysis

Means and standard deviations summarized the levels of cytokines and cell counts with and without M.tb infection and *PM_2.5_* exposure. Natural log transformations of counts were used when testing for the effects of exposure and stimulant on the counts in each scenario. These variance-stabilizing transformations created responses that were normally distributed and had roughly similar variances across stimulants and/or exposures. Because of the repeated measures on subjects, effects of M.tb infection, stimulants (PPD, PHA and lipopolysaccharide [LPS]) and *PM_2.5_* exposure on all outcomes (TNF-α, IL-10, CD69 and IFN-γ) were examined using mixed linear models with random effects for subject. Initially, we examined whether *PM_2.5_* exposure modified the effect of stimulants. Then, we examined the main effect of stimulants when there was no exposure and the main effect of *PM_2.5_* exposures within each level of stimulant. In the latter case, analyses were stratified by stimulant. F-tests were used to test the significance of the interactions and main effects. Although statistical models for inference were conducted using log-transformed values, the original values are plotted for ease of interpretation. For T-bet, because the responses were standardized to the case with no exposure/no stimulant, effects of stimulant among cases with no *PM_2.5_* exposures were assessed by comparing the fold change (relative to no stimulant) to 1. Specifically, the mean of the log-transformed responses was compared to zero using single-sample t-tests. Then, to assess the effect of *PM_2.5_* exposures within the stimulants, stratified mixed models, like those used for CD69 and IFN-γ, were employed with F-tests for the main effect of *PM_2.5_* exposure.

## 3. Results

### 3.1. Uptake of PM_2.5_ and M.tb in PBMC 

Initial interactions between M.tb and host immune cells involve phagocytosis of the bacteria by monocytes, macrophages and dendritic cells [23]. We studied the cellular uptake of M.tb and *PM_2.5_* in M.tb-infected and *PM_2.5_*-exposed PBMC by TEM. Following pre-exposure to 5 µg/mL of *PM_2.5_* for 20 h and subsequent infection with M.tb MOI 5 for 18 h, multiple M.tb bacteria were observed within membrane-enclosed vesicles of cells with monocyte morphology (Figure 1A,B,E,F). We also observed clusters of free, non-membrane bound, *PM_2.5_* in the cytoplasm of these cells (Figure 1C,D). *PM_2.5_* was noted extracellularly (circular insets in Figure 1E,F) and intracellularly concurrently with M.tb (Figure 1E,F). No PM clusters were noted in *PM_2.5_*-unexposed monocytes (not shown), in monocytes infected with M.tb only (Figure 1A,B), or within cell nuclei. 

### 3.2. Cytotoxic Effects of PM_2.5_ in Human PBMC 

We assessed if *PM_2.5_* exposure alters the viability of PBMC by measuring LDH leakage into culture supernatants. PBMC incubated in complete culture medium alone served as negative controls at each of the study time points. Exposure of PBMC to 1 and 5 µg *PM_2.5_* for 0, 1, 4, or 7 days (time periods resembling the durations of the M.tb growth control experiments) did not increase LDH release significantly (Figure 2A). Cellular cytotoxicity was also assessed in PBMC that were pre-exposed to *PM_2.5_* (final concentrations of 0, 1, or 5 µg/mL) for 20 h and then infected with M.tb at MOI 1 for 0, 1, 4, and 7 days (Figure 2B). No statistically significant differences were observed between no-PM control, M.tb-infected, or *PM_2.5_*-pre-exposed (1 and 5 µg/mL) and M.tb-infected PBMC on days 0, 1, 4, and 7 (*n* = 10) (Figure 2B). 

### 3.3. PM_2.5_ Exposure Effects on Apoptosis of Mononuclear Cells 

We also examined if *PM_2.5_* exposures of PBMC induce apoptosis in monocytes and lymphocytes at 24 and 48 h by flow cytometry. The 24 and 48 h time points were chosen to correspond with the cell culture periods during the subsequent immune response experiments. Exposure of PBMC to 5 µg *PM_2.5_* for 24 and 48 h did not cause any significant increases in apoptosis (annexin V-positive and propidium iodide-positive) in monocytes (Appendix A) or lymphocytes (Appendix A) compared to these cell subpopulations in PBMC exposed to 1 µg *PM_2.5_* or unexposed PBMC. We also assessed the effect of *PM_2.5_* pre-exposure (20 h) followed by M.tb infection at MOI 1 and 5 (18 h) on the induction of apoptosis in PBMC. No statistically significant differences in proportions of monocytes and lymphocytes undergoing apoptosis were observed between control PBMC and M.tb-infected PBMC at MOI 1 (Appendix A) and MOI 5 (Appendix A) or *PM_2.5_*-pre-exposed plus M.tb-infected PBMC at MOI 1 (Appendix A) and MOI 5 (Appendix A). 

### 3.4. PM_2.5_ Exposure Decreases Intracellular Growth Control of M.tb by PBMC 

One of the main protective immune effector functions of cytotoxic T cells during M.tb infection is the lysis of infected target cells thereby contributing to the control of intracellular growth of M.tb [24]. To assess whether *PM_2.5_* exposure alters the intracellular growth control of M.tb, PBMC were pre-exposed to *PM_2.5_* (final concentrations of 0, 1, and 5 µg/mL) for 20 h followed by infection with M.tb MOI 1 for 0 (2 hours), and 1, 4 and 7 days and CFU assays performed. M.tb CFU numbers were significantly higher in PBMC pre-exposed to 1 µg/mL of *PM_2.5_* on days 4 and 7 (Figure 3) (*p* < 0.05) and in PBMC pre-exposed to 5 µg/mL of *PM_2.5_* on days 1, 4, and 7 than in *PM_2.5_*-unexposed M.tb-infected PBMC. These observations indicate loss of intracellular growth control of M.tb by PBMC upon *PM_2.5_* exposure. Interestingly, while the observed *PM_2.5_*-induced loss of growth control of M.tb was dose-independent on days 1 and 4, M.tb cfu numbers were found to be significantly higher in PBMC pre-exposed to 5 µg/mL compared to 1 µg/mL of *PM_2.5_* (*p* < 0.05) on day 7. No significant differences in M.tb uptake (CFU numbers) were noted on day 0 (2 h after infection) at either of the *PM_2.5_* concentrations. 

### 3.5. PM_2.5_ Exposure Downregulates the Expression of CD69 on T Cells 

Expression of early activation marker CD69 on T cells during M.tb infection is a reliable measure of T cell activation [25] and critical for M.tb host immunity [26]. We evaluated the effect of *PM_2.5_* exposure on the expression of CD69 in PBMC T cell subsets on M.tb infection and PPD stimulation by flow cytometry. M.tb MOI 1 and MOI 5 infection, as well as PPD stimulation, but not *PM_2.5_* exposures, significantly increased CD69 expression in all PBMC T cell subsets (Figure 4A–C) compared to control PBMC. However, *PM_2.5_* 5 µg/mL pre-exposure of PBMC inhibited M.tb MOI 5-induced CD69 expression in CD3^+^, CD4^+^, and CD8^+^ T cells (Figure 4B) in PBMC (*p* < 0.01). Importantly, these *PM_2.5_* exposure effects were not a result of *PM_2.5_*-induced changes in the viability of the T cells (described above in Annexin-V FITC/propidium iodide double-staining experiments), thus suggesting that *PM_2.5_* exposures suppress early T cell activation processes during M.tb infection.

We also assessed the expression of an additional T cell activation marker, CD25 (the alpha chain of the IL-2 receptor), in *PM_2.5_* exposed M.tb-infected T cell subsets. Interestingly, CD25, which was constitutively expressed on T cell subsets (on 20% of CD3^+^, on 28% of CD4^+^, and on 5% of CD8^+^ T cells), did not show any significant changes upon M.tb infection and following *PM_2.5_* exposure (data not shown). 

### 3.6. PM_2.5_ Exposure Decreases M.tb-induced IFN-γ Expression in PBMC

IFN-γ is a key Th1 type cytokine required for protective human immunity against M.tb [8]. We determined the effect of *PM_2.5_* exposure on the expression of IFN-γ in CD3^+^, CD4^+^, CD8^+^, and CD16^+^ cells in M.tb-infected PBMC by flow cytometry. M.tb infections at MOI 1 and MOI 5 significantly induced IFN-γ expression in CD3^+^, CD4^+^, CD8^+^CD16^+^ (*p* < 0.001) and CD8^+^ (*p* = 0.002) T cells (Table 1). *PM_2.5_* pre-exposure reduced the expression of IFN-γ in CD3^+^ and CD4^+^ T cells of M.tb-infected PBMC and in CD8^+^CD16^+^ cells (*p* ≤ 0.05) as shown in Table 2. Together, the data described above show that PM exposure decreases the expression of IFN-γ by T-cells in response to M.tb. 

### 3.7. PM_2.5_ Exposure Decreases M.tb-Induced TNF-α Expression in CD3^+^ and CD14^+^ PBMC 

TNF-α is required for cell activation and inhibition of mycobacterial growth [4]. To explore if PM exposures modify TNF-α production in peripheral CD3^+^ and CD14^+^ cells, PBMC were pre-exposed to *PM_2.5_* (5 µg/mL) for 20 h and then infected with M.tb MOI 1 and MOI 5, or stimulated with PPD (10 µg/mL), phytohemagglutinin (PHA, 5 µg/mL; a potent T cell activator used as a positive control for T cell activation) or LPS (100 ng/mL), a potent NF-kB activator used as a positive control for monocyte activation, for an additional 8 h. Intracellular TNF-α expression in CD3^+^ T cells and CD14^+^ monocytes was then assessed by flow cytometry. TNF-α expression increased significantly upon infection with M.tb MOI 1 and MOI 5 or stimulation with PPD, PHA, and LPS in CD3^+^ T cells (Figure 5A) and in CD14^+^ cells (Figure 5B) compared to uninfected and no-PM control PBMC. As expected, TNF-α levels were significantly higher in CD14^+^ cells than CD3^+^ T cells, and significantly higher after stimulation with PHA in CD3^+^ T cells and significantly higher after stimulation with LPS in CD14^+^ cells (*p* < 0.05) (Figure 5A,B). 

Upon *PM_2.5_* exposure, the expression of TNF-α in response to M.tb MOI 1 and 5 infection or stimulation with PPD, PHA and LPS was significantly decreased in CD3^+^ and CD14^+^ PBMC compared to CD3^+^ and CD14^+^ PBMC infected with M.tb, or stimulated with PPD, PHA or LPS alone (*p* < 0.05) (Figure 5A,B). Although the PPD-induced TNF-α expression in CD3^+^ T cells upon *PM_2.5_* exposures was decreased in each of the six experiments, in the aggregate of six experiments PPD-induced TNF-α expression was not significantly reduced upon *PM_2.5_* exposure (Figure 5A, *p* = 0.08). Combined, these data show that *PM_2.5_* exposure decreases the expression of TNF-α in both CD3^+^ and CD14^+^ cells in PBMC infected with M.tb or stimulated with PPD, LPS or PHA. 

### 3.8. PM_2.5_ Exposure Induces the Production of IL-10 in M.tb-Infected PBMC

We examined the effect of *PM_2.5_* exposure on the production of IL-10 in CD3^+^ and CD14^+^ PBMC following M.tb infection or stimulation with PPD, PHA, or LPS by flow cytometry (Figure 6A,B). As expected, IL-10 expression levels were generally higher in CD14^+^ than in CD3^+^ cells. Interestingly, upon exposure to *PM_2.5_* alone, IL-10 expression in CD3^+^ (Figure 6A) and CD14^+^ cells (Figure 6B) significantly increased compared to *PM_2.5_* unexposed PBMC and PBMC that were M.tb-infected or PPD-, PHA-, and LPS-stimulated alone. IL-10 expression levels were higher in *PM_2.5_*-exposed- and M.tb-infected or PPD-, PHA-, and LPS-stimulated PBMC compared to PBMC infected with M.tb or stimulated with PPD, PHA, or LPS alone (Figure 6A,B). 

In contrast to the *PM_2.5_* exposure-mediated downregulation of M.tb-induced IFN-γ and TNF-α production, *PM_2.5_* exposure upregulates the expression of IL-10 in CD3^+^ cells. As IL-10 produced by T cells during M.tb infection contributes most to increased host susceptibility [27] and suppresses immune responses to TB [28], our findings indicate that *PM_2.5_* exposure promotes the anti-inflammatory capacity of T cells and thus the observed reduced expression of IFN-γ and TNF-α described. 

### 3.9. PM_2.5_ Exposure Reduces the Expression of Transcription Factor T-bet 

T-bet expression was assessed in PBMC at the same experimental conditions employed for the assessment of *PM_2.5_* effects on M.tb-induced IFN-γ production. The expression of T-bet was significantly increased in M.tb-infected PBMC compared to uninfected or unstimulated control PBMC (Figure 7A,B). While the expression of T-bet was slightly increased in PBMC exposed to *PM_2.5_* alone, T-bet expression was significantly reduced in M.tb-infected and PPD-stimulated PBMC upon *PM_2.5_* pre-exposure (*p* < 0.05) (Figure 7B). This observation is consistent with the observed decreased IFN-γ expression in CD3^+^ PBMC in which IFN-γ expression increased upon infection with M.tb alone (Table 1), but decreased upon *PM_2.5_* pre-exposure (Table 2). The observed decrease in M.tb-induced IFN-γ expression in CD3^+^ PBMC upon *PM_2.5_*-exposure may, at least in part, be a result of the reduced expression of T-bet upon PBMC exposure to *PM_2.5_*.

## 4. Discussion

Epidemiological evidence of significantly increased risk of TB development in cigarette smokers [29] and household/indoor [30] or urban air pollution-exposed persons [31] suggests that air pollutant exposures suppress antimicrobial immune effector functions. We have shown earlier that air pollution *PM_2.5_* and PM_10_ modifies innate antimycobacterial immune responses in human respiratory epithelial cells [20] and in human bronchoalveolar cells [21] and that exposure to diesel exhaust particles (DEP) alters the NFκB and IRF pathway target gene expression on M.tb infection in PBMC [19]. 

*PM_2.5_* deposits deep into the bronchoalveolar spaces on inhalation and can translocate into the circulatory system to exert adverse systemic health effects [32]. Evidence of translocation of ultrafine particles on inhalation of residential black carbon into the circulation was shown in a recent study in which urinary black carbon was found in children from Flanders, Belgium [33]. In another study, equally proving evidence of systemic effects on inhalational *PM_2.5_* exposure, significant associations were observed in traffic policemen between ambient *PM_2.5_* levels and changes in systemic inflammatory marker high sensitivity-CRP and immune markers (IgA, IgM, IgG, IgE, CD8 T cells) [34]. 

Studies assessing the effects of urban PM exposure on human T-cell responses have been lacking to date. To the best of our knowledge, this is the first report of suggested mechanisms underlying urban air pollution *PM_2.5_*-related susceptibility to M.tb infection by elucidating the effects of in vitro *PM_2.5_* exposure on human blood T cell immune responses to M.tb. In a first step, assessing cellular uptake and localization of *PM_2.5_* and M.tb in human peripheral blood monocytes by TEM, we observed clusters of free, non-membrane-bound PM in the cytoplasms of *PM_2.5_*-exposed monocytes (as well as extracellularly), and co-uptake of *PM_2.5_* and M.tb. These observations coincide with our earlier findings of co-uptake of DEP and M.tb in the cytoplasm of peripheral CD14^+^CD3^−^ monocytes. While we did not observe membrane or vacuole formations around DEP [19], or for that matter in the current study around PM inclusions, others observed a cytoplasmic localization of DEP in membrane-surrounded vesicles of human T cells [35] or in vacuoles of monocyte-derived macrophages [36]. Differences in cell types and/or particle sizes and compositions may underlie variations in cellular uptake mechanisms and intracellular localization of PM. 

An important assessment of the efficiency of antimycobacterial host immunity is the capacity of host cells to control the intracellular growth of M.tb. Studies in murine models have shown direct correlations between DEP exposures and higher lung loads of BCG (Bacille Calmette Guérin) in C57Bl/6J female mice [37], a reduced capacity of murine alveolar macrophages to kill M.tb, and suppressed expression of pro-inflammatory cytokines in lung tissues of female BALB/c mice upon experimental diesel exhaust inhalation [38]. 

In the current study, *PM_2.5_* exposure resulted in loss of intracellular growth control of M.tb by PBMC. As cell death from toxicity of *PM_2.5_* had to be considered as a contributor to the observed reduced capacity of PBMC to control the growth of M.tb, we assessed if *PM_2.5_* exposure alters cellular viability. Using multiple methods (trypan blue exclusion, LDH release by ELISA, apoptosis by flow cytometry) we could not identify cellular toxicity, neither in monocytes nor in lymphocytes between days 1 and 7 at the PM concentrations used. The observed *PM_2.5_* effects on M.tb growth control thus are due to functional impairments, not due to cell death. Findings from our studies in human bronchoalveolar cells further indicate that phagocytosis of M.tb is not significantly altered by PM-exposure of alveolar macrophages [21], thus also excluding the possibility that the observed *PM_2.5_*-mediated host immune modulations were due to differences in the initial cellular M.tb uptake. Taken together, our findings strongly suggest that the *PM_2.5_*-induced decrease in the capacity of PBMC to control M.tb growth is a consequence of defects in cellular effector mechanisms required for the killing of M.tb. 

Given the essential roles of activated CD4^+^ and CD8^+^ T cells in protective immunity against M.tb [2], we assessed the effect of exposure to PM on the M.tb-induced CD69 expression in CD3^+^, CD4^+^, and CD8^+^ PBMC. CD69 is a glycoprotein that is expressed on the surfaces of activated antigen-specific T cells [39] and serves as a costimulatory molecule for T-cell activation and proliferation [40]. Our findings indicate that *PM_2.5_* exposure downregulates M.tb-induced CD69 expression pointing to a potential additional *PM_2.5_*-mediated suppressive effect on antimycobacterial immunity. Further, as CD69 interacts with S1P receptor (S1PR_1_), a G protein-coupled receptor that is a target of the lipid signaling molecule Sphingosine-1-phosphate (S1P), reduced CD69 expression may also affect peripheral T cell retention and memory T cell formation [41]. The decreases in CD69 expression observed here contrast findings in another study in which DEP exposure did not affect CD69 expression in isolated human peripheral CD4^+^ and CD8^+^ T cells [35]. It is possible that differences in T cell activation processes (we used PBMC, allowing for T cell-monocyte interactions) and/or the differences in the chemical composition of *PM_2.5_* and DEP explain these dissimilar results. 

It was beyond the scope of this study to explore the impact of the chemical composition, the size ranges and the seasonal source of the *PM_2.5_* used here on M.tb-induced T cell responses. There is indeed evidence that PM size or seasonal composition differences affect biological responses [42]. Studies from our group [43,44] have shown that PM size and the composition related to both, the location and the season of its sampling, differentially affect exposure-related cellular cytotoxicity and modulation of immune responses to M.tb.

Many antibacterial cell effector functions are mediated by phagocytes that are activated by T cell-derived cytokines, in particular, IFN-γ and TNF-α [8]. We observed that urban *PM_2.5_* exposure decreases M.tb-induced IFN-γ production in CD3^+^ and CD4^+^ T cells (*p* < 0.05), findings that are consistent with observations of decreased IFN-γ release from LPS-activated T cells in murine models [45] and decreased IFN-γ release from M.tb-infected or PPD-stimulated human PBMC [19] on experimental DEP exposure. The reduced expression of IFN-γ in M.tb-infected CD8^+^CD16^+^ PBMC (Table 2) on *PM_2.5_* exposure in our study, further suggests suppression of cytotoxic responses to M.tb, as both CD8^+^ T cells [46] and NK cells [47] contribute to antimycobacterial killing mechanisms. Similarly, *PM_2.5_* exposure decreased TNF-α expression in both CD3^+^ and CD14^+^ M.tb-infected PBMC as well as in PPD-, PHA-, and LPS-stimulated PBMC (*p* < 0.05), providing further evidence that PM exposure decreases the production of protective human host pro-inflammatory cytokines during M.tb infection.

T-bet, a T-cell-associated transcription factor, is known to directly activate the expression of IFN-γ and required for the generation of Th1 immunity [48]. Mice lacking T-bet are susceptible to virulent M.tb infection with increased systemic bacterial burden, diminished IFN-γ production and increased IL-10 production [49]. IL-10, a product of both T cells and monocytes [50], inhibits immune responses to M.tb such as the production of pro-inflammatory cytokines [27]. IL-10 is increased in lung cells obtained by induced sputum from patients with active TB and increased IL-10 levels suppress effective host immune responses supporting the survival of mycobacteria [51]. Interestingly, in the current study, *PM_2.5_* exposure decreased the expression of T-bet in T cells and increased the production of IL-10 in T cells and monocytes in M.tb-infected PBMC. Therefore, we speculate that the *PM_2.5_*-induced loss of intracellular growth control in the current study may in part have been due to decreases in M.tb-induced IFN-γ production that was mediated by a reduced expression of T-bet and increased production of IL-10. It is interesting in this context that cigarette smoke, although very different from urban *PM_2.5_* in its chemical and particulate composition, has been shown to reduce T-bet expression during infection with M.tb in murine lung T cells [52]. 

In summary, our findings indicate that exposure to urban air pollution *PM_2.5_* alters T-cell responses to M.tb by (1) decreasing M.tb-induced surface expression of early T cell activation marker CD69, (2) inhibiting M.tb-induced intracellular expression of both pro-inflammatory IFN-γ and TNF-α, (3) decreasing M.tb growth control, (4) increasing the expression of anti-inflammatory cytokine IL-10, and (5) downregulating the expression of T-bet. Exposure to ‘real-world’-derived urban *PM_2.5_* thus simultaneously affects multiple M.tb-specific human host T cell functions as shown in our hypothetical model diagram (Appendix A). 

Considering our earlier and the current findings of suppression of vital antimycobacterial human immune cell functions following *PM_2.5_* exposures in vitro, one may speculate that PM exposures in highly air polluted environments may also adversely affect the efficacy of TB vaccines designed to induce T-cell-mediated immune responses [53]. PM-induced alterations of M.tb-specific T cell functions may also affect precision of diagnostic interferon gamma release assays (IGRA).

## 5. Conclusions

In conclusion, urban air pollution PM exposure mitigates the expression of protective antimycobacterial human host immune cell responses raising additional concerns about the adverse impact of air pollution on global TB control efforts. 

## Figures and Tables

**Figure 1 ijerph-16-04112-f001:**
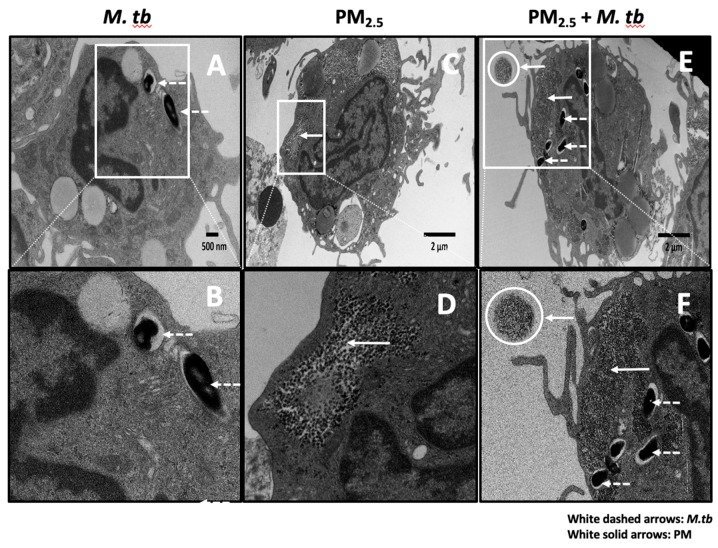
Transmission electron microscopy (TEM) of *PM_2.5_* and *Mycobacterium tuberculosis* (M.tb) uptake in human peripheral blood monocytes. (**A**,**B**) Endocytic vacuoles containing M.tb (white dashed arrows) showing uptake of multiple M.tb by a monocyte. (**C**,**D**) Monocyte containing air pollution *PM_2.5_* (after 20 h exposure to 5 µg/mL *PM_2.5_*, white solid arrows). (**E**,**F**) Monocyte showing clusters of *PM_2.5_* (white solid arrows, after 20 h exposure to 5 µg/mL *PM_2.5_*), and endocytic vacuoles containing M.tb (white dashed arrows, MOI 5) after M.tb infection for an additional 18 h. A free aggregate of *PM_2.5_* (white circles) is visible extracellularly in panels **E** and **F**. Panels **B**, **D**, and **F** are zoomed images of panels **A**, **C**, and **E** respectively.

**Figure 2 ijerph-16-04112-f002:**
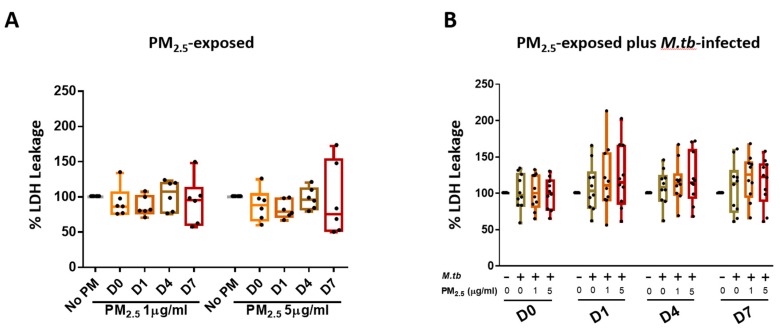
Cytotoxic effects of *PM_2.5_* in human peripheral blood mononuclear cells (PBMC). (**A**) PBMC from six study subjects were exposed to *PM_2.5_* at final concentrations of 0, 1, and 5 µg/mL for 0, 1, 4, and 7 days. As a measure of cytotoxicity, leakage of lactate dehydrogenase (LDH) into culture supernatants was determined. (**B**) Cytotoxic effects of *PM_2.5_* on PBMC from ten study subjects pre-exposed to *PM_2.5_* for 20 h followed by infection with M.tb at multiplicities of infection (MOI) 1 for 0 (2 hours), 1, 4, and 7 days were determined by measuring LDH concentrations in culture supernatants. Percent LDH expression is expressed as 5/95 percentile box-and-whiskers where the center represents the 50th percentile, the upper hinge the 75th percentile, and the lower hinge the 25th percentile.

**Figure 3 ijerph-16-04112-f003:**
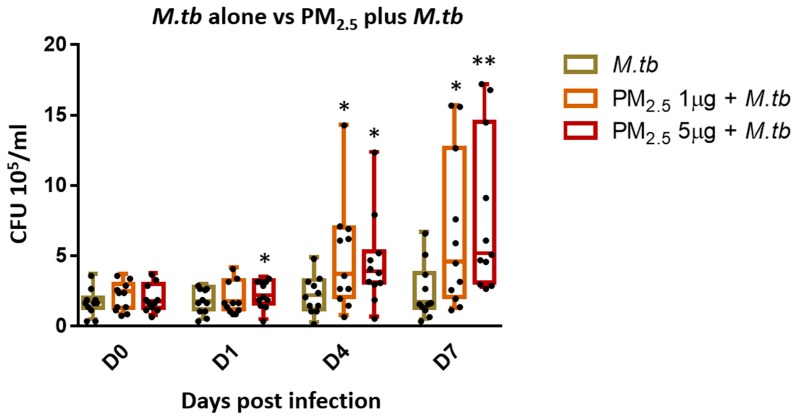
*PM_2.5_* exposure causes loss of intracellular growth control of M.tb by PBMC. PBMC from 11 study subjects were pre-exposed to *PM_2.5_* (final concentrations of 0, 1, and 5 µg/mL) for 20 h followed by infection with M.tb MOI 1 for 0 (2 h), 1, 4, and 7 days. CFU assays were performed to determine the effects of *PM_2.5_* exposure on M.tb growth control of PBMC. CFU numbers from 11 independent experiments are expressed as 5/95 percentile box-and-whiskers where the center represents the 50th percentile, the upper hinge the 75th percentile, and the lower hinge the 25th percentile. Statistically significant differences between results of PBMC infected with M.tb alone, and *PM_2.5_*-exposed and M.tb-infected PBMC are shown with single (*p* < 0.05) or double (*p* < 0.01) asterisks.

**Figure 4 ijerph-16-04112-f004:**
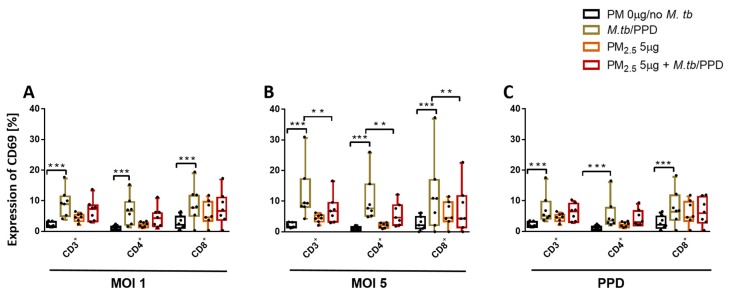
*PM_2.5_* exposure downregulates the surface expression of the early T cell activation marker CD69. (**A**,**B**) PBMC from seven study subjects were pre-exposed to *PM_2.5_* (5 µg/mL) for 20 h followed by infection with M.tb MOI 1 and 5 for 18 h. Flow cytometric analysis of CD69 expression, PBMC were surface stained with anti-CD3, -CD4, -CD8, -CD69 monoclonal antibodies and viability dye eFluor780. Percent expression of CD69 in seven independent experiments on CD3^+^, CD4^+^, and CD8^+^ cells is shown with 5/95 percentile box-and-whiskers where the center represents the 50th percentile, the upper hinge the 75th percentile, and the lower hinge the 25th percentile. Statistically significant increases between expression levels of M.tb-uninfected and M.tb-infected PBMC are shown with triple (*p* < 0.001) asterisks while statistically significant decreases between results for M.tb-infected and *PM_2.5_*-exposed/M.tb-infected PBMC are shown with double (*p* < 0.01) asterisks.

**Figure 5 ijerph-16-04112-f005:**
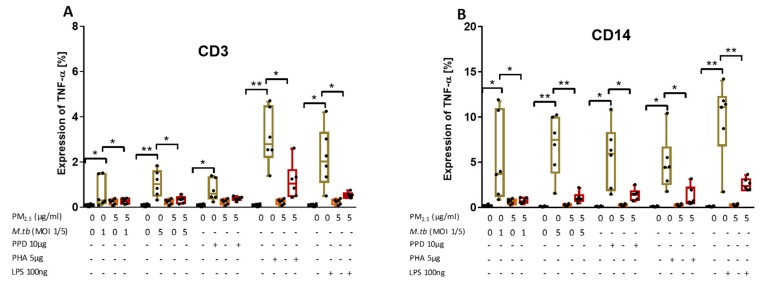
*PM_2.5_* exposure decreases M.tb-induced TNF-α expression in PBMC. (**A**,**B**) PBMC from six study subjects were pre-exposed to *PM_2.5_* (5 µg/mL) for 20 h followed by infection with M.tb MOI 1 and 5 or stimulation with PPD (10 µg/mL), PHA (5 µg/mL) or LPS (100 ng/mL) for 8 h. Surface staining with anti-CD3 and anti-CD14 monoclonal antibodies was performed followed by permeabilization and intracellular staining for detection of TNF-α by flow cytometry. The values represent the percentage of CD3^+^ and CD14^+^ PBMC expressing TNF-α in six independent experiments as 5/95 percentile box-and-whiskers where the center represents the 50th percentile, the upper hinge the 75th percentile, and the lower hinge the 25th percentile. Statistically significant increases between proportions of cells expressing TNF-α in uninfected and M.tb-infected or PPD-, PHA-, and LPS-stimulated PBMC are shown with single (*p* < 0.05) or double (*p* < 0.01) asterisks while statistically significant decreases between results for M.tb-infected or PPD-, PHA-, and LPS-stimulated and *PM_2.5_*-exposed and M.tb-infected or PPD-, PHA-, and LPS-stimulated PBMC are shown with single (*p* < 0.05) or double (*p* < 0.01) asterisks.

**Figure 6 ijerph-16-04112-f006:**
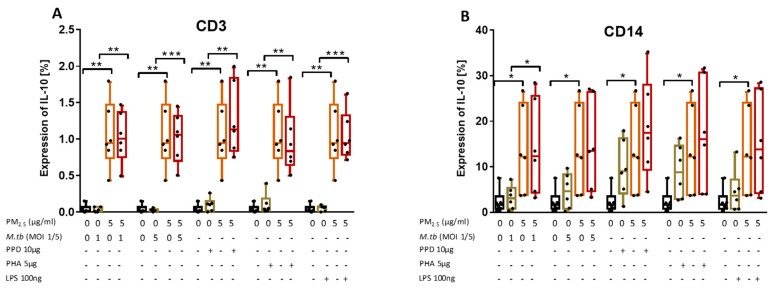
*PM_2.5_* exposure increases IL-10 expression in PBMC. (**A**,**B**) PBMC from six study subjects were pre-exposed to *PM_2.5_* (5 µg/mL) for 20 h followed by infection with M.tb MOI 1 and 5 or stimulation with purified protein derivative (PPD) (10 µg/mL), phytohemagglutinin (PHA) (5 µg/mL) or LPS (100 ng/mL) for 18 h. Surface staining with anti-CD3 and anti-CD14 monoclonal antibodies was performed followed by permeabilization and intracellular staining for detection of IL-10 by flow cytometry. The values represent the percentage of CD3^+^ and CD14^+^ PBMC expressing of IL-10, in six independent experiments, as 5/95 percentile box-and-whiskers where the center represents the 50th percentile, the upper hinge the 75th percentile, and the lower hinge the 25th percentile. Statistically significant increases between proportions of cells expressing IL-10 in uninfected and *PM_2.5_*-exposed PBMC are shown with single *(p* < 0.05) or double (*p* < 0.01) asterisks while statistically significant increases between results for M.tb-infected and PM-exposed-M.tb-infected PBMC are shown with single (*p* < 0.05), double *(p* < 0.01) or triple (*p* < 0.001) asterisks.

**Figure 7 ijerph-16-04112-f007:**
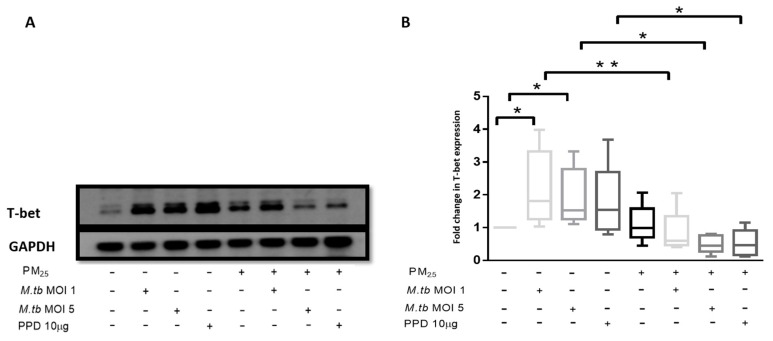
*PM_2.5_* exposure decreases T-bet expression in PBMC. (**A**) Representative western blot results showing the effects of *PM_2.5_*-exposure on the expression of T-bet. PBMC were pre-exposed to *PM_2.5_* (5 µg/mL) for 20 h followed by infection with M.tb MOI 1 or MOI 5 or stimulation with PPD (10 µg/mL) for 18 h. PBMC were then lysed, cellular protein extracts prepared and resolved by SDS-PAGE, and western blotting performed with specific antibodies for T-bet and GAPDH (housekeeping gene). (**B**) Densitometric results of five independent experiments were obtained and normalized for GAPDH. Data represent fold change in T-bet expression expressed as 5/95 percentile box-and-whiskers where the center represents the 50th percentile, the upper hinge the 75th percentile, and the lower hinge the 25th percentile. Statistically significant increases between T-bet expression levels of uninfected and M.tb-infected PBMC are shown with single (*p* < 0.05) while statistically significant decreases in T-bet expression levels between M.tb-infected and *PM_2.5_*-exposed/M.tb-infected PBMC are shown with single (*p* < 0.05) and double (*p* < 0.01) asterisks.

**Table 1 ijerph-16-04112-t001:** Increases in IFN-γ expression levels in T-cell subsets of PBMC infected with M.tb MOI 1 and MOI 5 compared to uninfected and unstimulated (no PM ) PBMC.

T Cell Subset	M.tb MOI	*p*-Value vs. Uninfected, Unstimulated (No *PM_2.5_*) PBMC
CD3^+^	1	0.0001
CD3^+^	5	<0.0001
CD4^+^	1	0.0005
CD4^+^	5	<0.0001
CD8^+^	1	0.0026
CD8^+^	5	0.0002
CD8^+^CD16^+^	1	0.0014
CD8^+^CD16^+^	5	0.0002

IFN-γ expression levels in T-cell subsets of PBMC infected with M.tb. PBMC from nine study subjects were infected with M.tb MOI 1 and 5 for 18 h. Surface staining with monoclonal antibodies against CD3, CD4, CD8, CD16, and viability dye eFluor780 was performed followed by permeabilization and intracellular staining with anti-IFN-γ antibodies for flow cytometry analysis. p-values show significant increases in IFN-γ expression levels in T cell subsets upon infection with M.tb MOI 1 and MOI 5 compared with uninfected PBMC.

**Table 2 ijerph-16-04112-t002:** *PM_2.5_* exposure reduces M.tb-induced IFN-γ expression in T-cell subsets of PBMC.

T Sell Subset	PM_2.5_ Pre-Exposure Followed by M.tb Infection	*p*-Value vs. MOI1 and MOI5 Combined
CD3+	*PM_2.5_* + M.tb	0.0318
CD4+	*PM_2.5_* + M.tb	0.0268
CD8+	*PM_2.5_* + M.tb	0.0814
CD8+CD16+	*PM_2.5_* + M.tb	0.0522

*PM_2.5_* exposure and M.tb-induced IFN-γ expression in T-cell subsets of PBMC. PBMC from nine study subjects were pre-exposed to *PM_2.5_* (5 µg/mL) for 20 h followed by infection with M.tb MOI 1 and 5 for 18 h. Surface staining with monoclonal antibodies against CD3, CD4, CD8, CD16, and viability dye eFluor780 was performed followed by permeabilization and intracellular staining with anti-IFN-γ antibodies for flow cytometry analysis. p-values show significant *PM_2.5_*-induced decreases in IFN-γ expression levels in T cell subsets infected with M.tb for a combined analysis of MOI1 and MOI5.

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
