# Peer review of "Urban Air Pollution Particulates Suppress Human T-Cell Responses to Mycobacterium Tuberculosis"

_ijerph, 2019, doi:10.3390/ijerph16214112_

Round 1
Reviewer 1 Report
This is a very valuable and well-prepared paper that clearly showing the studied effect of ambient PM2.5 exposure on T cells in human peripheral blood mononuclear cells. The research team had prior experiences and important findings in their previous studies on other sources like diesel PM2.5. The experiment is well conducted and controlled. The manuscript is also well organized and clearly present. I’d like to suggest accepting this after a minor edit.
Exposure concentration. In the present experiment, 2 levels of 1 and 5 ug/mL were adopted in exposure, besides a control of no particles. Can the author discuss or relate this to average daily exposure dose for the general population (maybe in Mexico)? Lab mechanism(s) studies sometimes have much higher exposure levels compared to that in real world, which is understandable as at low exposure levels it would be difficult to study the impacts, however, this often raises another problem is that the levels are too high and may less likely happen for the general population. PM characteristics. As also mentioned by the authors on lines 525-527, particle size and composition profiles affect the biological effects as well. I do agree that this may be beyond the scope of this study, but giving important roles of size and compositions in determining biological response, it is suggested to add briefly particle chemical compositions and size information, possibly from past results in the same project and in a supporting table for figure. This will be also helpful in generalizing the results and in future comparison studies by others. Lines 481-483, any past studies on relationship between PM Exposure and tuberculosis in other cells? Suggest improving resolutions of figures 2-5.Author Response
Please see attachment with replies to reviewers.

Reviewer 2 Report
Lines 101 to 113: Is there knowledge about the composition and morphology of the particles? Can that have influence on the results of this study?
Author Response
"Please see attachment".
